# Acute Administration of Exogenous Lactate Increases Carbohydrate Metabolism during Exercise in Mice

**DOI:** 10.3390/metabo11080553

**Published:** 2021-08-21

**Authors:** Inkwon Jang, Jisu Kim, Sunghwan Kyun, Deunsol Hwang, Kiwon Lim

**Affiliations:** 1Department of Sports Medicine and Science in Graduate School, Konkuk University, Gwangjin-gu, Seoul 05029, Korea; inkwon555@konkuk.ac.kr (I.J.); kimpro@konkuk.ac.kr (J.K.); y10345@konkuk.ac.kr (S.K.); hds49@konkuk.ac.kr (D.H.); 2Physical Activity and Performance Institute (PAPI), Konkuk University, Gwangjin-gu, Seoul 05029, Korea; 3Department of Physical Education, Konkuk University, Gwangjin-gu, Seoul 05029, Korea

**Keywords:** exogenous lactate, supplement, exercise, metabolism, carbohydrate oxidation

## Abstract

In this study, we investigated the effects of exogenous lactate administration before exercise on energy substrate utilization during exercise. Mice were divided into exercise control (EX) and exercise with lactate intake (EXLA) groups; saline/lactate was administered 30 min before exercise. Respiratory gas was measured during moderate intensity treadmill exercise (30 min). Immediately after exercise, blood, liver, and skeletal muscle samples were collected and mRNA levels of energy metabolism-related and metabolic factors were analyzed. At 16–30 min of exercise, the respiratory exchange ratio (*p* = 0.045) and carbohydrate oxidation level (*p* = 0.014) were significantly higher in the EXLA than in the EX group. Immediately after exercise, the muscle and liver glycogen content and blood glucose level of the EXLA group were lower than those of the EX group. In addition, muscle mRNA levels of *HK2* (hexokinase 2; *p* = 0.009), a carbohydrate oxidation-related factor, were higher in the EXLA than in the EX group, whereas the expression of *PDK4* (pyruvate dehydrogenase kinase 4; *p* = 0.001), *CS* (citrate synthase; *p* = 0.045), and *CD36* (cluster of differentiation 36; *p* = 0.002), factors related to oxidative metabolism, was higher in the EX than in the EXLA group. These results suggest that lactate can be used in various research fields to promote carbohydrate metabolism.

## 1. Introduction

Supplement consumption before exercise is an important factor that directly affects metabolism during exercise, and many studies have been conducted on this topic [1,2]. Consumption of supplements, such as capsaicin [3], caffeine [4], and carnitine [2,5], is known to improve muscular endurance performance by increasing the energy supply through fat oxidation during exercise [2,3,4,5]. However, the use of supplements is associated with side effects, such as excessive sensitivity and stomachache [4,5,6]. Therefore, it is necessary to discover new exercise supplements with minimal side effects.

In earlier studies, intracellular acidosis in skeletal muscles was identified along with a high accumulation of lactate when high intensity exercise was continued until exhaustion. Accordingly, lactate was regarded a “fatigue-inducing molecule” and “metabolic end-product” of an anaerobic process [7,8,9]. However, further research on lactate confirmed that the amount of lactate produced during exercise is not sufficient to have a direct effect on acidosis [10]. In high-intensity exercise, fatigue is caused by the depletion of stored glycogen and intracellular acidosis in skeletal muscles develops because of an increase in the H^+^ ion concentration produced from ATP hydrolysis [10,11,12]. Furthermore, it is known that lactate is the main energy source of the body [13,14,15] and is a gluconeogenic precursor in the liver [15,16,17]. It is transported to oxidative fibers through the blood by monocarboxylate transport proteins and used as an energy source [18,19]. Hence, lactate is considered an important molecule for metabolic regulation.

Recent studies have investigated the effects of exogenous lactate intake on energy metabolism. Yu Kitaoka et al. [20] suggested that lactate, as a signaling molecule, upregulates genes related to mitochondrial function by confirming that the expression of *PGC*-1α, *PDK4*, and *UCP3* was upregulated in the skeletal muscle of mice 3 h after lactate administration. Takahashi et al. [21] reported that exogenous lactate intake before exercise for 4 weeks increases the activity of mitochondrial cytochrome c oxidase in the oxidative phenotype of the muscle. Hoshino et al. [22] reported that chronic post-exercise lactate administration increased the levels of monocarboxylate transporter 1 (MCT1) and improved skeletal muscle glycogen recovery and storage capacity. Moreover, Kyun et al. [23] reported that exogenous lactate administration increased the mRNA and protein levels of protein-synthesis-related factors in the skeletal muscle, suggesting the potential of lactate as a supplement that promotes muscle synthesis. In addition, at rest, exogenous lactate intake increased glycogen synthesis- and fat metabolism-related factors [24]. These findings suggest the potential of exogenous lactate as a new exercise supplement.

Nevertheless, the effect of lactate intake on metabolism during exercise has not yet been directly investigated. In the aforementioned study by Kyun et al. [24], it was also confirmed that glycogen synthesis- and fat metabolism-related factors increased over time after lactate administration. This study investigated the metabolic changes in exogenous lactate intake at rest over time; however, it did not determine the effect of lactate on metabolism during exercise. Russ et al. [25] reported that lactate intake did not improve aerobic capacity, such as the oxygen uptake (VO_2_) peak and time to exhaustion. Hence, it is essential to directly identify the energy substrate changes that take place during exercise to clarify the effect of lactate on metabolic substrate use and to define its value as an exercise supplement. Therefore, we aimed to investigate the effect of exogenous lactate intake on energy metabolism and substrate use during exercise and to evaluate its potential as an exercise supplement.

## 2. Results

### 2.1. Energy Metabolism during Exercise

There was no significant difference in metabolism during the total exercise period (30 min) between the EX (exercise) and EXLA (exercise + lactate intake) groups. However, we observed (Figure 1a,c, Figure 2a, Figure 3a, Figure 4a) that the lines representing the data of the two groups in the metabolic graphs crossed at 15 min. Based on this observation, we divided the 30 min exercise period into the following two 15 min periods: 0–15 min (0–15 M) and 16–30 min (16–30 M). The results showed no significant differences between the two groups in the 0–15 M period. In the 16–30 M period, VO_2_ and fat oxidation (FO) were not significantly different between the two groups (Figure 1b,d); however, significantly higher carbon dioxide production (VCO_2_; *p* = 0.027), respiratory exchange rate (RER; *p* = 0.045), and carbohydrate oxidation (CO; *p* = 0.014) were observed in the EXLA group than in the EX group (Figure 2d, Figure 3d, Figure 4d). Although not statistically significant, the total energy expenditure (EE) of the EXLA group tended to be higher than that of the EX group in the 16–30 M period (Figure 5). Thus, lactate intake before exercise may increase the use of carbohydrates as an energy substrate during exercise.

### 2.2. Glycogen Content

We analyzed the glycogen content in the liver and skeletal muscle to determine how the intake of lactate before exercise affects the use of glycogen, a storage form of carbohydrate, during exercise. The results showed that the muscle glycogen content (Figure 6a) was significantly lower in the EXLA group than in the EX group (*p* = 0.008). The liver glycogen content (Figure 6b) was also significantly lower in the EXLA group than in the EX group (*p* = 0.001). These results indicate that lactate intake before exercise can increase the use of glycogen as an energy substrate during exercise.

### 2.3. Blood Analysis-Whole Blood

Blood analysis was performed immediately after exercise using whole blood to identify factors directly related to the use of energy substrates during exercise (Table 1). The results showed that the concentrations of glucose (*p* = 0.001) and triglyceride (TG; *p* = 0.001) in the EXLA group were significantly lower than those in the EX group. Although not statistically significant, the concentration of lactate in the EXLA group was higher than that in the EX group (*p* = 0.061).

### 2.4. Blood Analysis-Serum

The concentration of glycerol and free fatty acids (FFAs) was measured in the serum (Table 2). According to the results, the concentrations of glycerol (*p* = 0.004) and FFAs (*p* = 0.003) of the EXLA group were significantly lower than those of the EX group.

### 2.5. mRNA Expression of Energy Metabolism-Related Factors

To determine how the administration of lactate before exercise affects the use of energy substrates during exercise, we examined the expression of energy metabolism-related mRNAs in the gastrocnemius skeletal muscle (Figure 7). The expression level of hexokinase 2 (*HK2*) was significantly higher in the EXLA group than in the EX group (*p* = 0.009). The mRNA levels of pyruvate dehydrogenase kinase 4 (*PDK4*; *p* = 0.001) and citrate synthase (*CS*; *p* = 0.045) were significantly lower in the EXLA group than in the EX group. In addition, the levels of pyruvate carboxylase (*PC*; *p* = 0.041), glucose 6-phosphatase (*G6Pase*; *p* = 0.014), and monocarboxylate transporter 1 (*MCT1*; *p* = 0.03) were significantly higher in the EXLA group than in the EX group. Furthermore, the levels of cluster of differentiation 36 (*CD36*) were significantly lower in the EXLA group than in the EX group (*p* = 0.002).

## 3. Discussion

The purpose of this study was to investigate the effect of exogenous lactate administration as an exercise supplement on the use of energy substrates during exercise. The results showed that exogenous lactate intake did not differ in the total exercise period of 30 min. However, CO increased in the 16–30 M period, and the glycogen contents and blood glucose concentration in the liver and muscle were low immediately after exercise. In addition, mRNA expression associated with carbohydrate utilization was upregulated. 

In general, when the exercise intensity is high, VO_2_, VCO_2_, RER, CO, and EE increase and FO decreases. This is considered a result of an increased rate of carbohydrate oxidation, which quickly provides energy to meet the increased energy requirements, and a decreased rate of fat oxidation, which is a slower process [26,27]. However, the results of breathing gas generated during exercise confirmed that the VCO_2_, CO, and RER of the EXLA group were significantly higher than those of the EX group at 15 min of exercise. Although there was no change in exercise intensity and no significant difference between the two groups in VO_2_ and EE after 16 min, lactate intake might have increased the carbohydrate oxidation rate and the use of carbohydrates as an energy substrate during exercise. 

Carbohydrates and fats are the main energy sources used by the human body. Carbohydrates are stored in the form of glucose in the blood and as glycogen in the liver and muscles and are immediately supplied as an energy source when required [28,29]. By analyzing the blood glucose concentration immediately after exercise and the glycogen content stored in the liver and muscles, we found that the blood glucose concentration and glycogen contents of the EXLA group were significantly lower than those of the EX group. Most likely, the energy requirements were met through the supply of blood glucose and glycogen when the ratio of carbohydrates used during exercise increased. 

Brooks et al. [30] reported that a high blood lactate concentration inhibits adipose tissue lipolysis. In a previous study, the blood lactate concentration was found to increase rapidly at 15 min of exercise [20,31]. Our blood analysis results confirmed that the levels of the lipid metabolism indicators TG, glycerol, and FFAs were significantly lower in the EXLA group than in the EX group. However, we did not confirm the blood lactate concentration after lactate ingestion. Nonetheless, based on our results, it is expected that lactate intake before exercise increases the concentration of blood lactate, thereby inhibiting adipose tissue lipolysis and thus fat metabolism during exercise. 

To confirm the effect of lactate intake before exercise on energy substrate use during exercise, respiratory gas, blood glucose, and glycogen contents were measured. The findings demonstrated that carbohydrate metabolism during exercise increased when lactate was consumed before exercise. To confirm these results, mRNA expression analysis related to energy metabolism was performed. The results showed that the expression of *HK2* was significantly higher in the EXLA group than in the EX group. HK2 is a major enzyme that converts glucose to glucose-6-phosphate and regulates the overall speed of glycolysis [32,33]. Hence, an increased expression level of *HK2* indicates high levels of carbohydrate oxidation. Further, the levels of factors related to oxidative energy metabolism or metabolic utilization, such as *PDK4* and *CS*, were significantly lower in the EXLA group than in the EX group. PDK4 inhibits the oxidation of carbohydrates through phosphorylation and inactivation of the pyruvate dehydrogenase complex that converts pyruvate to acetyl Co-A [28]. CS converts acetyl Co-A and oxaloacetate into citrate in the first step of the tricarboxylic acid cycle [34]. CS is thus used as an indicator of oxidative metabolic capacity [35]. 

MCT1, a lactate carrier that promotes the use of lactate in skeletal muscles [36], plays a role in inducing lactate removal by transferring lactate transported through the blood to the mitochondria [19]. According to previous studies, chronic lactate intake after exercise increases the expression level of MCT1 protein, leading to effective removal of increased lactate during exercise [22]. Our results showed that the EXLA group had significantly higher *MCT1* mRNA expression level than the EX group, suggesting that the expression level of *MCT1* was increased to remove a large amount of lactate. In addition, CD36 plays a role in transporting fat to muscle tissue to oxidize fat; it is highly correlated with whole body fat oxidation or utilization [37]. Spriet et al. reported that CD36 deficiency was related to problems with fat transport and oxidation [38]. Our *CD36* mRNA expression analysis showed significantly lower *CD36* mRNA levels in the EXLA group than in the EX group. This is consistent with the blood TG, glycerol, and FFA results shown in Table 1 and Table 2, indicating that the fat metabolism of the EXLA group during exercise was not optimal. 

PC, a key factor, is considered a rate-limiting enzyme in the gluconeogenesis process. PC determines whether pyruvate is converted to oxaloacetate or oxidized to acetyl-CoA. In addition, *G6Pase* converts glucose-6-phosphate to glucose and is considered to be a regulatory factor for gluconeogenesis and glycogen metabolism [39]. Our results showed that the EXLA group had significantly higher *PC* and *G6Pase* mRNA levels than the EX group. These results suggest that both glycolysis and gluconeogenesis were upregulated in the EXLA group than in the EX group. When the respiratory gas data, blood glucose levels, liver and muscle glycogen contents, and *HK2* expression at the mRNA level were examined, the activation of glycolysis in the EXLA group was significant. Furthermore, since there are only limited studies on the activation of gluconeogenesis in the EXLA group, it is necessary to approach it from various perspectives, and further studies should be conducted. Based on the low expression of *PDK4* and *CS* mRNA and low blood TG, FFA, and glycerol levels in the EXLA group, we estimated that the oxidative metabolic response in this group was lower than that in the EX group. It was hypothesized that the activation of gluconeogenesis to resynthesize carbohydrates immediately after exercise was higher in the EXLA group than in the EX group due to glycogen depletion and low blood glucose levels.

Interestingly, we have previously confirmed the potential of lactate as a supplement by examining the effects of lactate intake at rest on metabolism [23,24]. The main aim of the present study was to investigate the effect of lactate intake on metabolism during exercise. Therefore, we did not use sedentary mice as the control. However, to clarify the direct effect of lactate and the synergistic effects of lactate administration in combination with exercise on energy metabolism in detail, it would be important to include a resting group to the present experimental setting or to extend the duration of the experimental period.

Nevertheless, we investigated that the lactate intake before exercise increased the use of carbohydrates as energy substrates during exercise. Therefore, this study highlights the potential of using lactate as a supplement in various physiological condition that require the promotion of carbohydrate metabolism.

## 4. Materials and Methods

### 4.1. Animal Care

Seven-week-old male ICR mice (*n* = 16), purchased from Orient Bio Inc. (Seongnam, Korea), were used in this study, which was approved by the Konkuk University Institutional Animal Care and Use Committee (No. KU19149). Prior to the initiation of this study, the mice were adapted to the laboratory environment for a week. The mice were housed in standard plastic cages (four mice per cage) under controlled humidity (45–50%), temperature (22 ± 1 °C), and lighting (12:12-h light-dark cycle; lights on at 07:00 a.m.) conditions [40], and fed ad libitum with a standard commercial diet (60% carbohydrate, 20% protein, and 9.6% fat). 

### 4.2. Study Design

Mice were randomly divided into the EX and EXLA groups (*n* = 8 per group, average weight: EX group = 32.4 ± 0.84, EXLA group = 32.5 ± 1.28). Prior to conducting the experiment, all groups were fasted for 8 h. The EXLA group was administered 3 g/kg of sodium lactate with an oral sonde and the EX group an equal amount of a saline solution [23,41]. The mice were orally administered the appropriate supplement 30 min before a moderate intensity (about 60–70% VO_2max_) exercise was conducted for 30 min (treadmill exercise at a speed of 18 m/min and slope of 6°) [42,43]. Immediately after exercise, the mice were anesthetized with 10 µL/g of 1.25% avertin and the tissues were collected.

### 4.3. Metabolic Analysis during Exercise

To investigate how lactate intake affects the use of energy substrate during exercise, experiments were conducted using metabolic analysis machines. The mice were orally administered lactate 30 min before the measurement. The mice used a metabolic treadmill chamber, which was divided one by one, to measure energy metabolism during exercise (treadmill exercise at a speed of 18 m/min and slope of 6°). The metabolic chambers were constructed using an open-circuit method and the volume of each chamber was 3 L. The average flow rate for each chamber was set to 3 L/min. An acrylic tube was connected to each chamber for air volume manipulation. Respiratory gas (O2 uptake and CO2 production) was analyzed using a mass spectrometer (model ARCO-2000, ARCO System, Chiba, Japan) and switching system (model ARCO-2000-GS-8, ARCO System) allowing the spectrometer to sample the gas from each chamber. VO2, VCO2, RER, CO, FO, and EE were calculated based on the measured respiratory gas.

### 4.4. Determination of Blood Parameters

Venous blood samples were collected immediately after exercise, and blood analysis was performed using whole blood and serum. To collect the serum, the venous blood samples were allowed to clot at 24–25 °C for 30 min, centrifuged at 2000× *g* for 15 min at 4 °C, and transferred to new tubes before being stored at −80 °C. Whole blood was used to measure the concentration of blood glucose (ACCU CHEK Performa Glucometer, Roche, Diagnostics, Penzberg, Germany), lactate (Lactate Pro2, LT-1730, ARKRAY, Kyoto, Japan), and TG (Standard LipidoCare Strips–Lipid Profile, 02LA10G, SD Biosensor, Suwon, Korea). Serum was used in ELISA kits for the analysis of the concentration of glycerol (EGLY-200, BioAssay System, Hayward, CA, USA) and FFAs (K612-100, BioVision, Milpitas, CA, USA).

### 4.5. Glycogen Content Analysis

To determine the glycogen content of each tissue, we collected the liver and plantaris muscle immediately after exercise and stored them at −80 °C until analysis. Ten milligrams of liver and muscle were used for the analysis. The glycogen contents were measured using the Glycogen Assay Kit (K646-100, BioVision, Milpitas, CA, USA) according to manufacturer’s instructions.

### 4.6. Reverse Transcription-Polymerase Chain Reaction (RT-PCR) Analysis

We performed RT-PCR to determine the mRNA levels of metabolism-related factors. The mRNA obtained from the gastrocnemius muscle was used to measure the expression levels of *GAPDH* and of five other factors. Total RNA was extracted from the gastrocnemius muscle using the QIAzol Lysis Reagent (79306; Qiagen, Hilden, Germany). We synthesized complementary DNA (cDNA) from total RNA using the amfiRivert cDNA Synthesis Platinum Master Mix (R5600; GenDEPOT, Katy, TX, USA). We then performed the RT-reaction using the following protocol: annealing for 5 min at 25 °C, extension for 50 min at 42 °C, and RT inactivation for 15 min at 70 °C. For RT-PCR, cDNA was amplified using the amfiEco Taq DNA polymerase (P0701; GenDEPOT) and the following primer pairs: *HK2*-F and -R, *PDK4*-F and -R, *CS*-F and -R,* MCT1*-F and -R, *CD36*-F and -R, *PC*-F and -R, and *G6Pase*-F and -R (Table 3). The cycling conditions were as follows: an initial denaturation for 2 min at 94 °C, followed by 18–30 cycles of 15 s at 94 °C, 30 s at 50–60 °C (primer Tm °C), and 1 min at 72 °C. Finally, we separated the products using 1% agarose gels and visualized them using the Safe-Pinky DNA gel-staining solution (S1001-025; GenDEPOT).

### 4.7. Statistical Analysis

All data were analyzed using IBM SPSS Statistics 25 software (Armonk, NY, USA). Significant differences in the means were determined using an independent sample *t*-test. Significant differences in values over time were determined using a two-way repeated analysis of variance (ANOVA). A *p*-value of <0.05 was considered statistically significant. The results are presented as the mean ± standard deviation (SD).

## Figures and Tables

**Figure 1 metabolites-11-00553-f001:**
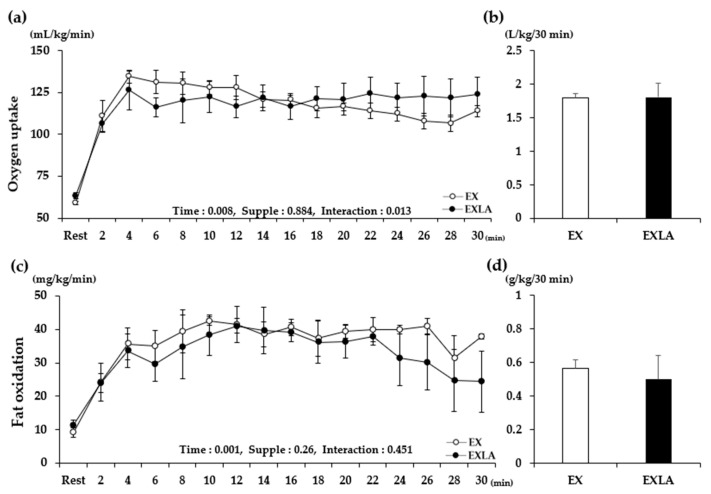
Changes in VO_2_ and FO during exercise. (**a**,**c**) VO_2_ and FO changes over time for 30 min, respectively; (**b**,**d**) total VO_2_ and FO for 30 min, respectively. Supple, supplement effect; Inter, interaction effect; VO_2_, oxygen uptake; FO, fat oxidation; EX, exercise control group; EXLA, exercise with lactate intake group. Values represent the mean ± standard deviation (*n* = 8).

**Figure 2 metabolites-11-00553-f002:**
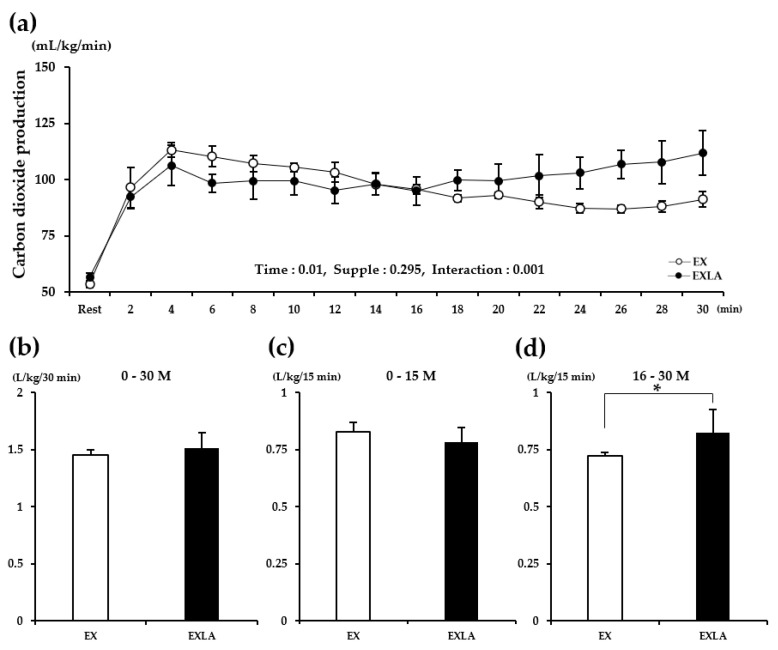
Changes in VCO_2_ during exercise. (**a**) VCO_2_ changes over time for 30 min; (**b**) total VCO_2_ for 30 min; (**c**,**d**) total VCO_2_ for 0–15 min (0–15 M) and 16–30 min (16–30 M), respectively. Supple, supplement effect; Inter, interaction effect; VCO_2_, carbon dioxide production; EX, exercise control group; EXLA, exercise with lactate intake group. Values represent the mean ± standard deviation (*n* = 8). * *p* < 0.05.

**Figure 3 metabolites-11-00553-f003:**
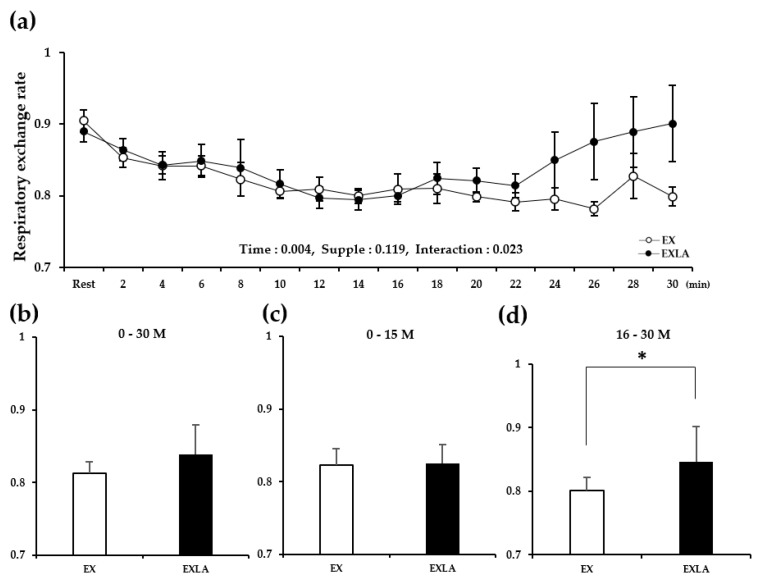
Changes in RER during exercise. (**a**) RER changes over time for 30 min; (**b**) average RER for 30 min; (**c**,**d**) average RER for 0–15 min (0–15 M) and 16–30 min (16–30 M), respectively. Supple, supplement effect; Inter, interaction effect; RER, respiratory exchange rate; EX, exercise control group; EXLA, exercise with lactate intake group. Values represent the mean ± standard deviation (*n* = 8). * *p* < 0.05.

**Figure 4 metabolites-11-00553-f004:**
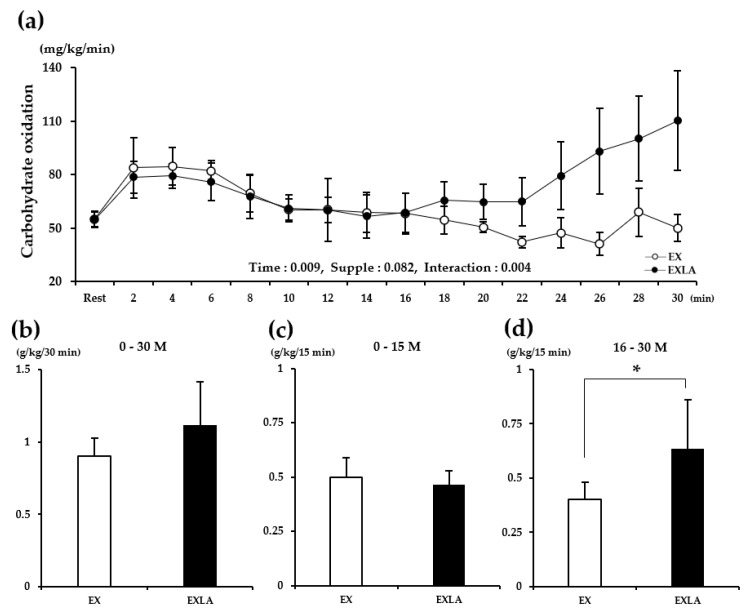
Changes in CO during exercise: (**a**) CO changes over time for 30 min; (**b**) total CO for 30 min; (**c**,**d**) total CO for 0–15 min (0–15 M) and 16–30 min (16–30 M), respectively. Supple, supplement effect; Inter, interaction effect; CO, carbohydrate oxidation; EX, exercise control group; EXLA, exercise with lactate intake group. Values represent the mean ± standard deviation (*n* = 8). * *p* < 0.05.

**Figure 5 metabolites-11-00553-f005:**
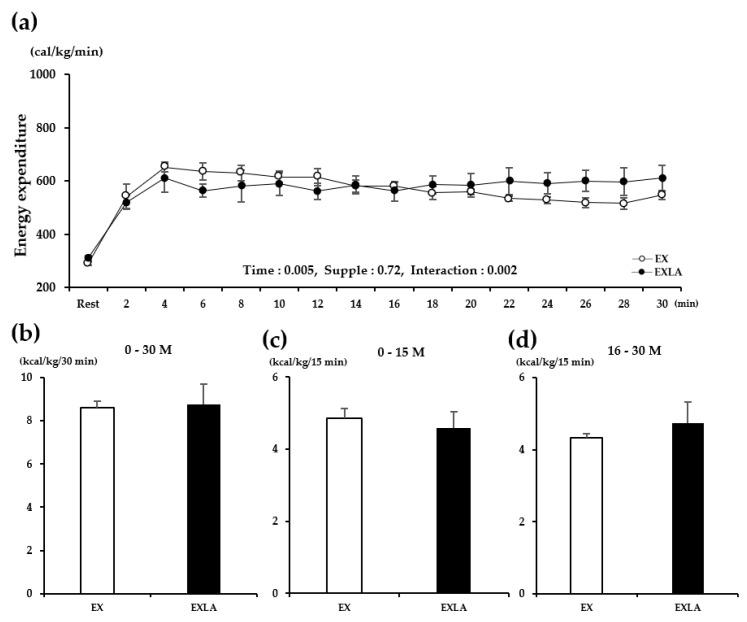
Changes in EE during exercise. (**a**) EE changes over time for 30 min; (**b**) total EE for 30 min; (**c**,**d**) total EE for 0–15 min (0–15 M) and 16–30 min (16–30 M), respectively. Supple, supplement effect; Inter, interaction effect; EE, energy expenditure; EX, exercise control group; EXLA, exercise with lactate intake group. Values represent the mean ± standard deviation (*n* = 8).

**Figure 6 metabolites-11-00553-f006:**
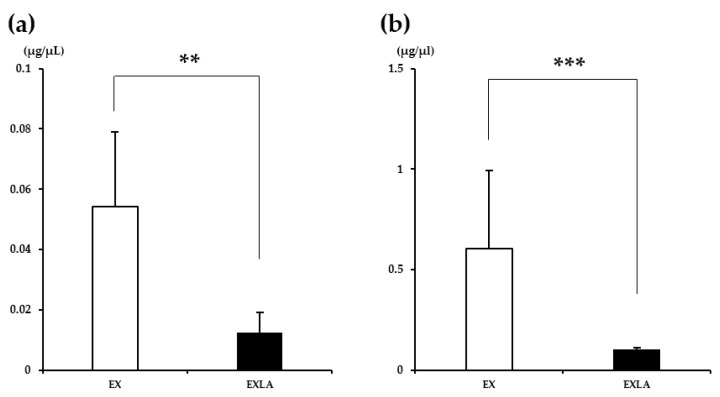
Glycogen content in the liver and muscle after exercise. (**a**) Muscle glycogen content; (**b**) liver glycogen content. EX, exercise control group; EXLA, exercise with lactate intake group. Values represent the mean ± standard deviation (*n* = 8). ** *p* < 0.01, *** *p* < 0.001.

**Figure 7 metabolites-11-00553-f007:**
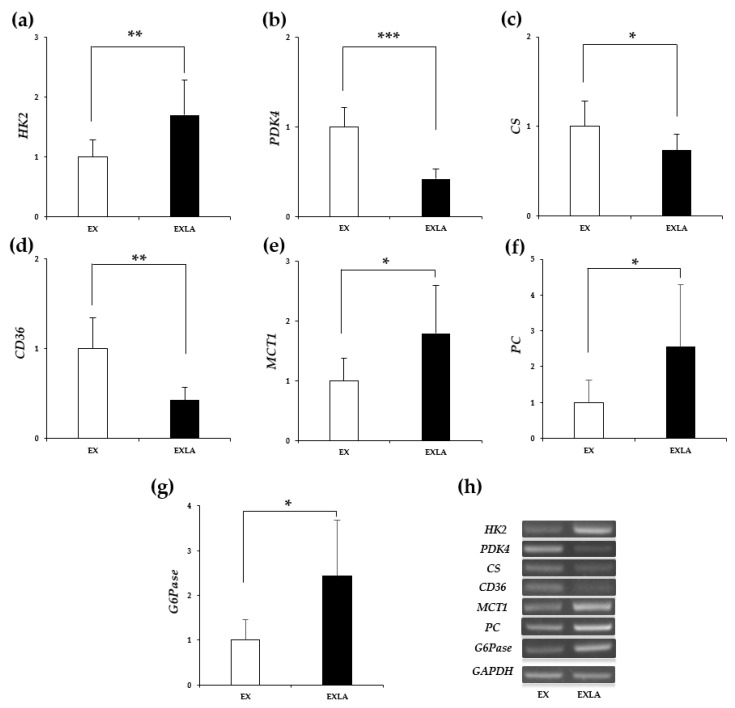
Expression of energy metabolism-related mRNAs in the gastrocnemius muscle: (**a**–**g**) quantification of *HK2*, *PDK4*, *CS**, CD36*, *MCT1, PC,* and *G6Pase* mRNA levels, respectively; (**h**) representative gene expression images. *HK2*, hexokinase 2; *PDK4*, pyruvate dehydrogenase kinase 4; *CS*, citrate synthase; *CD36*, cluster of differentiation 36; *MCT1*, monocarboxylate transporter 1; *PC*, pyruvate carboxylase; *G6Pase*, glucose 6-phosphatase; *GAPDH*, glyceraldehyde 3-phosphate dehydrogenase; EX, exercise control group; EXLA, exercise with lactate intake group. *GAPDH* was used for the normalization of the target mRNA expression. Values represent the mean ± standard deviation (*n* = 8). * *p* < 0.05; ** *p* < 0.01; *** *p* < 0.001.

**Table 1 metabolites-11-00553-t001:** Whole blood glucose, triglyceride, and lactate concentrations.

	EX	EXLA
Glucose (mg/dL)	158.5 ± 26	65.5 ± 27.2 ***
Triglyceride (mg/dL)	145.9 ± 13.3	92.3 ± 15.7 ***
Lactate (mmol/L)	7.4 ± 1.8	10.8 ± 4.2

EX, exercise control group; EXLA, exercise with intake lactate group. Values represent the mean ± standard deviation (*n* = 8). *** *p* < 0.001.

**Table 2 metabolites-11-00553-t002:** Serum glycerol and free fatty acid concentrations.

	EX	EXLA
Glycerol (mmol/L)	0.51 ± 0.05	0.4 ± 0.08 **
Free fatty acid (mmol/L)	0.59 ± 0.16	0.37 ± 0.07 **

EX, exercise control group; EXLA, exercise with intake lactate group. Values represent the mean ± standard deviation (*n* = 8). ** *p* < 0.01.

**Table 3 metabolites-11-00553-t003:** Primer sequences for reverse transcription-polymerase chain reaction (RT-PCR).

Gene	Sequence
*HK2*	F-5′ ATC GCC GGA TTG GAA CAG AA 3′
	R-5′CTC CGT GAA TAA GCA GGC GA 3′
*PDK4*	F-5′ CGC CTG GCC AAT ATC CTG AA 3′
	R-5′ GCC TTG AGC CAT TGT AGG GA 3′
*CS*	F-5′ CAA GTC ATC TAC GCC AGG GAC A 3′
	R-5′ CAA AGC GTC TCC AGC TAA CCA AG 3′
*CD36*	F-5′ GGC CAA GCT ATT GCG ACA T 3′
	R-5′ CAG ATC CGA ACA CAG CGT AGA 3′
*MCT1*	F-5′ GGC CTG AGC AAG TCA AGC TA 3′
	R-5′ GCA AAT CCA AAG ACT CCG GC 3′
*PC*	F-5′ GCA GCC TTT GGG AAT GGA 3′
	R-5′ GGT GAG ACG TGA GCG AAG TTG 3′
*G6Pase*	F-5′ CAG AAT GGG TCC ACC TTG ACA 3′
	R-5′ GGG CTT CAG AGA GTC AAA GAG ATG 3′
*GAPDH*	F-5′ AAC TTT GGC ATT GTG GAA GG 3′
	R-5′ ACA CAT TGG GGG TAG GAA CA 3′

*HK2*, hexokinase 2; *PDK4*, pyruvate dehydrogenase kinase 4; *CS,* citrate synthase; *CD36*, cluster of differentiation 36; *MCT1*, monocarboxylate transporter 1; *PC*, pyruvate carboxylase; *G6Pase*, glucose 6-phosphatase; *GAPDH*, glyceraldehyde 3-phosphate dehydrogenase.

## Data Availability

The data presented in this study are available on request from the corresponding author.

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
