# Peer review of "Acute Administration of Exogenous Lactate Increases Carbohydrate Metabolism during Exercise in Mice"

_metabolites, 2021, doi:10.3390/metabo11080553_

Round 1

Reviewer 1 Report

The manuscript of Jang et al. submitted to Metabolites for publication entitled  “Acute Administration of Exogenous Lactate Increases Carbo-2 hydrate Metabolism during Exercise in Mice”  describes the effect of exogenous lactate administration before, during, and after exercise on energy substrate utilization and concluded through experimentation that the exogenous lactate acts as an exercise supplement. The experiments are based on a sound footing by adopting the latest methodologies to reach to final conclusion. Results obtained are well illustrated with an appropriate discussion of old and latest citations. Statistical analysis suggests the study as significant. I, therefore, recommend its publication in its present form.

Author Response

Thank you for your thoughtful suggestions and insights for the publication of our manuscript in this journal.

Based on the ‘English language and style’ part you pointed out, the manuscript was reviewed and additionally re-edited in English. (The revised parts are marked with a yellow highlighter.)

I have attached the revised manuscript below, so please check it.

Thank you for your consideration.

Reviewer 2 Report

The authors studied interesting topic and whole project was well dome. The study deserves publication. However, for better understanding and interpretation data several especially methodological details are missing.

Methods:

Line 251 Body weight of mice should be mentioned in the article

Line  267 It is important to elucidate amount of carbohydrate fat and proteins eaten per g of bod weight in standard commercial diet by mice. This is important because reader should know how much extra energy is in sodium lactate given in dosage 3g/kg BW. This will help to differentiate pharmaceutical (buffer) and metabolic (nutritional) effect of sodium lactate.

Line 263 Method of peroral administration of lactate should be given (administration via tube of given to the diet)

Line 260 How long were mice fasting before experiment?

 Line 281 I suppose that mice were killed for muscle and liver sampling. The method of anaesthesia and killing is not described in the manuscript.    

Results:

To determine the glycogen loss in liver and muscle   pre-exercise data of glycogen amount in muscle and liver must be presented.

Author Response

We thank you for your thoughtful suggestions and insights. The manuscript has benefited from these insightful suggestions.

Based on the points you pointed out, the manuscript was reviewed and revised as a whole, with particular focus on detailing the ‘Methods’ and ‘Results’ part.

In addition, after editing the manuscript, it was re-edited in English. (The revised parts are marked with a yellow highlighter.)

I have attached the revised manuscript below, so please check it.

I hope that the answers I wrote are sufficient explanations.

Thank you for your consideration.

Reviewer 3 Report

Introduction

  1. The authors addressed the safety of choosing exercise supplements. However, guidance in sports nutrition clarified that under recommendation amount, adverse effects are tiny. Besides, lactate, same as exogenous ketone bodies, are relatively safe because they originate in the organisms.  Refs that might help: (1)Kesl, S. L., Poff, A. M., Ward, N. P., Fiorelli, T. N., Ari, C., Van Putten, A. J., ... & D’Agostino, D. P. (2016). Effects of exogenous ketone supplementation on blood ketone, glucose, triglyceride, and lipoprotein levels in Sprague–Dawley rats. Nutrition & metabolism13(1), 1-15.  (2) Ma, S., & Suzuki, K. (2019). Keto-adaptation and endurance exercise capacity, fatigue recovery, and exercise-induced muscle and organ damage prevention: a narrative review. Sports7(2), 40.
  2. Ref. 23, non-supplement groups are missing, we do not know whether exercise play role in the glucose-metabolism enhancement.
  3. Seldom previous studies about lactate supplements or endogenous supplements were introduced in the intro part. If there are others, they should all be introduced.

Study design

1. The authors claimed that mice were divided randomly into the ex or non-ex groups,

did they divide the mice based matched on weight or average weight?

2. Absolute control (non-ex, non-la) is missing.

3. The authors started the exercise test after 30min of lactate administration.

What is the basis?

4. The speed control is not described. Are the speeds automated adjusted during the experiment to match with the VO2?

5. What is the basis to choose the plantaris tissue for glycogen analysis?

6. Is an equivalent of 3g/kg lactate prescription achievable in humans?

Results

  1. Abbreviations must be addressed under each picture or table.
  2. Non-ex controls are not used, no pre-data are corrected neither. This raises questions.
  3. Gluconeogenesis-related genes are not measured.
  4. MCT1 also transports other molecules such as ketone body

Discussion and Conclusion

The authors claimed that lactate supplements may be useful, however, there are no baseline data to compare.

Not much expression mRNA during exercise was measured, therefore we could hardly conclude that lactate is useful or useless, or overdose danger. 

The author claimed lactate administration impacts fatty acid metabolism during exercise, however, key genes during fat accumulation or modulation are not measured.

Author Response

Thank you for your thoughtful suggestions and insights.

The manuscript has benefited from these insightful suggestions. I was able to think from various perspectives while revising the points you pointed out, and I think it was an opportunity to broaden my insights as a researcher.

In addition, after editing the manuscript, it was re-edited in English. (The revised parts are marked with a yellow highlighter.)

I have attached the revised manuscript below, so please check it.

I hope that the answers I wrote are sufficient explanations.

Thank you.

Round 2

Reviewer 2 Report

no comments

Author Response

Thank you again for allowing us to review and publish in a great journal.

Reviewer 3 Report

Thank you for revising the manuscript, the results are new to the scientific society, however, the conclusion must be rewrite- the link between the results presented in this study does not support or support only weakly that lactate administration will benefit T2D or obesity.  

Author Response

Thanks for the constructive review.

I completely agree with what you point out.

Therefore, it was written by excluding the parts including T2D or obesity that were pointed out in the manuscript.

It was written on lines 266-268 (The revised part is highlighted in yellow.).

"Therefore, this study highlights the potential of using lactate as a supplement in various physiological condition that require the promotion of carbohydrate metabolism."

Thank you again for allowing us to review and publish in a great journal.
